# Train Once, Deploy Anywhere: Edge-Guided Single-source Domain Generalization for Medical Image Segmentation

**Jun Jiang** [1]                                        JUNJIANG.STEVE@GMAIL.COM
[1] *Shenzhen Institute for Advanced Study, UESTC, Shenzhen, China*
**Shi Gu** [1,2]                                           GUS@UESTC.EDU.CN
[2] *School of Computer Science and Engineering, UESTC, Chengdu, China*

**Editors:** Accepted for publication at MIDL 2024

## Abstract

In medical image analysis, unsupervised domain adaptation models require retraining when receiving samples from a new data distribution, and multi-source domain generalization methods might be infeasible when there is only a single source domain. These will pose formidable obstacles to model deployment. To this end, we take the "Train Once, Deploy Anywhere" as our objective and consider a challenging but practical problem: Single-source Domain Generalization (SDG). Meanwhile, we note that (i) the medical image segmentation applications where generalization errors often come from imprecise predictions at the ambiguous boundary of anatomies and (ii) the edge of the image is domain-invariant, which can reduce the domain shift between the source and target domain in all network layers. Specifically, we borrow the prior knowledge from Digital Image Processing and take the edge of the image as input to enhance the model attention at the boundary of anatomies and improve the generalization performance on unknown target domains. Extensive experiments on three typical medical image segmentation datasets, which cover cross-sequence, cross-center, and cross-modality settings with various anatomical structures, demonstrate our method achieves superior generalization performance compared to the state-of-the-art SDG methods. The code is available at https://github.com/thinkdifferentor/EGSDG.

**Keywords:** Domain Generalization, Transfer Learning, Medical Image Segmentation.

## 1. Introduction

Medical image segmentation is a crucial task in clinical applications. In recent years, deep segmentation networks have achieved remarkable progress(Butoi et al., 2023; Isensee et al., 2021). However, when there is domain shift between the training and testing data, the performance of data-driven deep models degrades dramatically, like scanning technique, acquisition parameters, device manufacturers, etc. Recently, many efforts of Domain Generalization (DG) and Unsupervised Domain Adaptation (UDA) have been made to improve the model's generalization ability on the target domain (Su et al., 2023; Feng et al., 2023). Further, domain generalization can be divided into Multi-source Domain Generalization (MDG) and Single-source Domain Generalization (SDG). On the one hand, MDG models (Dou et al., 2019; Li et al., 2019) are designed with multiple source domains to learn domain-invariant representations, and it may not work when there is only a single source domain. On the other hand, the high cost of medical image labeling and the strict regulations of privacy protection make it difficult to obtain large amounts of medical data. Besides, previous UDA works (Feng et al., 2023; Chen et al., 2019) require retraining when

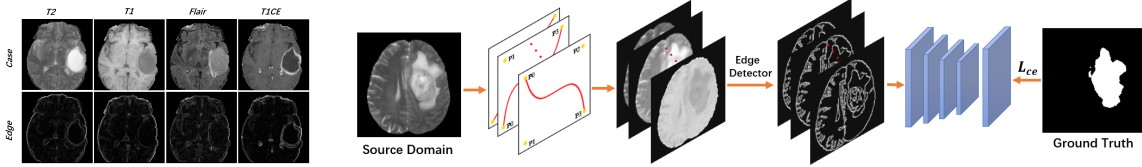

Figure 1: Example case with corresponding Sobel gradient map of BraTS'19, which can be used as domain-invariant information to guide the training process (left). Overview of our proposed *Edge-Guided Single-source Domain Generalization* (EGSDG) for medical image segmentation (right).

receiving the samples from a new data distribution, which leads to the high cost of model deployment. To this end, we take the "*Train Once, Deploy Anywhere*" as our objective and consider a challenging but practical setting: single-source domain generalization.

For segmentation tasks, models often make inaccurate predictions at target boundaries. This weakness is more pronounced for domain generalization or adaptation segmentation tasks in medical images due to the domain gap and ambiguous boundary of anatomies. Recently, some works have proposed corresponding solutions and made significant progress (Liu et al., 2022; You et al., 2023; Feng et al., 2023). However, there are several limitations to them. First, they do not directly take the image edge as the model input, which weakens the supervision of edge information during the training process. Second, they acquire the image edge or shape priors by learning, which will take lots of training time. Based on these insights, we employ the edge detection algorithm to get image edge maps and use them as input to train the model directly. This effectively filters out domain-specific information and significantly improves the generalization ability. The major contributions of this work are as follows: (i) We make a comprehensive analysis to the impact of image edge on the model generalization ability, including the positions of edge supervision signals, such as shallow, deep, or output layers; the fusion strategies of edge map and feature map, such as concatenating it with shallow or deep features and directly as network input. (ii) For the challenging yet essential SDG problem of medical image segmentation, we propose a simple yet effective approach EGSDG, which significantly improves the generalization performance on unknown target domains. (iii) We conducted extensive experiments on three typical medical image segmentation datasets that cover various anatomical structures. With only a single source domain, our method achieves superior generalization performance on the unknown target domain compared to the state-of-the-art SDG methods.

## 2. Related Works

### 2.1. SDG of Nature Image

SDG models of nature images can be divided into two mainstream methods: (1) *the image-level method*, which improve the model generalization by data augmentation with the help of existing large datasets (e.g. ImageNet) (Yue et al., 2019; Lee et al., 2022), and (2) *the feature-level method*, which aims to learn domain-invariant segmentation network by removing the style information of feature maps with normalization or whitening strategy

(Choi et al., 2021; Peng et al., 2022). However, these models may not work well on grayscale medical images, because there are significant differences in texture, structure, and data privacy policies between medical and natural images.

### 2.2. SDG of Medical Image

In medical images, there are fewer works on the SDG segmentation task compared to natural images. Most of these models (Liu et al., 2022; Ouyang et al., 2021; Su et al., 2023) conduct data augmentation on the source domain to improve the model's robustness. Different from the previous works, we introduce an edge-guided model, which filters the domain-specific information effectively and improves the generalization ability significantly.

### 2.3. Edge-Guided Methods

Recently, many efforts of edge-guided methods have been made to raise attention to the segmentation boundary and improve the generalization ability of models. Cardace et al. (Cardace et al., 2021) presented a novel low-level adaptation strategy with semantic edges and displacement maps from shallow features to obtain sharp predictions. CIConv (Lengyel et al., 2021) exploited a visual inductive prior derived from physics-based reflection models and cast a number of color-invariant edge detectors as trainable layers for domain adaptation. In contrast to existing methods, we utilize the edge detector to extract edge maps of images and take them as input to train the model directly.

## 3. Methodology

### 3.1. Preliminaries

Edge detectors significantly filter out useless information, while preserving the important structural properties of an image. There are a large number of edge detection algorithms available, each designed to be sensitive to specific types of edges like edge orientation, noise environment, and edge structure. We take the most classic ones for exploring, including Canny (Canny, 1986), AutoCanny (Rong et al., 2014), Roberts (Roberts, 1963), Prewitt (Prewitt et al., 1970), Sobel (Kittler, 1983), and Laplacian (LeCun et al., 1998). For an image, its gradient at $(x, y)$ is defined as vector $\bigtriangledown f(x, y)$, which is composed of the partial derivative of the image in the X and Y directions:

$$\bigtriangledown f(x, y) = [G_x, G_y] = \left[\frac{\partial f}{\partial x}, \frac{\partial f}{\partial y}\right] \tag{1}$$

The modulus and direction of the gradient are defined by:

$$|\bigtriangledown f(x, y)| = \sqrt{G_x^2 + G_y^2}, \theta(x, y) = \arctan(\frac{G_y}{G_x}) \tag{2}$$

For Laplacian, the second derivative is defined as:

$$\bigtriangledown^2 f(x, y) = \frac{\partial^2 f(x, y)}{\partial x^2} + \frac{\partial^2 f(x, y)}{\partial y^2} \tag{3}$$

Note that the digital images are discrete and different edge detection algorithms differ in the way of $G_x$ and $G_y$ calculation, which are provided in Appendix A. The details of Canny and AutoCanny algorithms can be found in Appendix B and (Rong et al., 2014). Compared to Canny algorithm, AutoCanny does not need to manually set Gaussian smoothing parameters and the double thresholds.

### 3.2. Problem Definition

In single-source domain generalization, we are given a single source domain $D^s = \{(x_i^s, y_i^s)\}_{i=1}^{N_s}$, where $s$ represents the domain ID, $x_i^s \in \mathbb{R}^{H \times W \times 3}$ is the $i$-th image in the source domain $s$. $y_i^s \in \mathbb{R}^{H \times W}$ is the corresponding ground truth mask, and $N_s$ is the total number of samples. Given unseen target domain $D^t = \{x_i^t, y_i^t\}_{i=1}^{N_t}$, which is not accessible during the training process, we aim to minimize the error between prediction $\hat{y}_i^t$ and ground truth mask $y_i^t$.

### 3.3. Edge-Guided SDG

Edge or gradient information is one of the most important features of an image. The edge of image is domain-invariant, which can reduce the domain shift between the source and target domain in all network layers (Lengyel et al., 2021). Different from previous works, TASD (Liu et al., 2022) establishes the dictionary learning to extract the explicit shape priors and CIConv (Lengyel et al., 2021) derived from the complex Kubelka-Munk theory to build a learnable edge detector layer, our model is borrowed from the classical edge detection algorithm with less computational complexity and more stable performance. Visualization comparison of classical edge detection algorithms and CIConv refer to Appendix C.

In addition, data augmentation can enrich the gradient information of the training samples, which will bring huge performance gains to our edge-guided model's generalization ability (Details refer to Appendix D). For medical images, we expect to map the source image to diverse grayscale value distribution and keep the appearance of the anatomic structures perceivable at the same time. Motivated by Model Genesis (Zhou et al., 2019), we employ the Bézier Curve (Mortenson, 1999) as our data augmentation method, which is generated from two end points ($P_0$ and $P_3$) and two control points ($P_1$ and $P_2$), defined as:

$$B(t) = \sum_{i=0}^{n} \binom{n}{i} P_i (1-t)^{n-i} t^i, n = 3, t \in [0, 1] \tag{4}$$

where $t$ is a fractional value along the length of the line.

The learning process of our EGSDG is shown in Figure 1. Firstly, we perform the BézierCurve data augmentation on source samples ($x_i^s \in \mathbb{R}^{H \times W \times 3}$) before the training stage. Then, the edge detector is exploited to extract the edge maps ($e_i^s \in \mathbb{R}^{H \times W}$) of the augmented samples. Finally, we take the edge maps $e_i^s$ as input to train the segmentation network $\phi^w$ with parameters $w$ by minimizing cross-entropy loss:

$$\mathcal{L}_{ce}(\phi^w; D^s) = -\sum_i [y_i^s, log(\phi^w(e_i^s))] \tag{5}$$

We use the edge detector to compress the image into a single-channel edge map. It effectively filters domain-specific information and trains a model with high generalization performance. The network locates the segmentation object by the gradient change (Roberts, Prewitt, Sobel, and Laplacian) or the anatomy contour (Canny and AutoCanny) of the target area.

Table 1: Quantitative comparison of different methods on BraTS'19 (left) and Prostate (right) datasets. Note that CIConv* indicates training with BézierCurve augmented dataset and the result of SADN is reported by that method on BraTS'18 dataset.

| Method | Source Domain: T2 | | | |
| --- | --- | --- | --- | --- |
| | T1 | T1ce | Flair | Avg. |
| Upper Bound | 74.42 | 71.64 | 82.75 | 76.27 |
| Lower Bound | 13.82 | 11.58 | 66.61 | 30.67 |
| IBN-Net | 34.37 | 48.27 | 42.33 | 41.66 |
| SW | 31.83 | 40.48 | 34.95 | 35.75 |
| RobustNet | 8.59 | 10.14 | 68.29 | 29.01 |
| SADN | 49.36 | 38.09 | 75.87 | 54.44 |
| CSDG | 46.76 | 44.99 | 60.20 | 50.65 |
| CIConv | 15.36 | 20.83 | 76.07 | 37.42 |
| CIConv* | 53.82 | 52.69 | 74.05 | 60.19 |
| EGSDG w/o Aug. | 51.38 | 50.35 | 71.63 | 57.79 |
| EGSDG w/ Aug. | 62.59 | 54.68 | 77.07 | 64.78 |

| Method | Source Domain: Site B | | | | | |
| --- | --- | --- | --- | --- | --- | --- |
| | Site A | Site C | Site D | Site E | Site F | Avg. |
| Upper Bound | 89.13 | 89.96 | 89.31 | 87.76 | 89.34 | 89.10 |
| Lower Bound | 63.62 | 19.42 | 81.06 | 83.89 | 71.17 | 63.83 |
| IBN-Net | 67.36 | 46.79 | 65.09 | 71.45 | 76.88 | 65.51 |
| SW | 70.83 | 51.71 | 70.89 | 51.96 | 68.97 | 62.87 |
| RobustNet | 73.27 | 55.04 | 77.41 | 54.79 | 70.21 | 66.14 |
| CSDG | 69.75 | 61.47 | 74.27 | 76.31 | 70.54 | 70.47 |
| CIConv | 73.48 | 63.51 | 80.80 | 62.15 | 74.93 | 70.97 |
| CIConv* | 76.41 | 59.74 | 76.63 | 78.10 | 77.17 | 73.61 |
| EGSDG w/o Aug. | 72.70 | 59.54 | 83.00 | 70.36 | 81.11 | 73.34 |
| EGSDG w/ Aug. | 78.51 | 64.16 | 82.95 | 77.34 | 78.20 | 76.23 |

Table 2: Quantitative comparison of different methods on MMWHS dataset. Note that CIConv* indicates training with the BézierCurve augmented dataset and the result of SADN is reported by that method.

| Method | Source Domain: MRI | | | | | Source Domain: CT | | | | |
| --- | --- | --- | --- | --- | --- | --- | --- | --- | --- | --- |
| | AA | LAC | LVC | MYO | Avg. | AA | LAC | LVC | MYO | Avg. |
| Upper Bound | 89.74 | 84.99 | 87.44 | 83.34 | 86.38 | 80.76 | 82.29 | 92.38 | 78.23 | 83.42 |
| Lower Bound | 32.18 | 35.92 | 19.53 | 9.42 | 24.26 | 18.44 | 8.84 | 38.72 | 9.65 | 18.91 |
| IBN-Net | 59.04 | 67.63 | 67.34 | 45.49 | 59.88 | 31.23 | 42.36 | 59.91 | 34.63 | 42.03 |
| SW | 52.94 | 69.52 | 64.28 | 44.64 | 57.84 | 38.95 | 47.62 | 62.82 | 33.30 | 45.67 |
| RobustNet | 68.07 | 74.68 | 62.56 | 46.09 | 62.85 | 52.27 | 60.08 | 67.26 | 32.97 | 53.14 |
| SADN | 51.42 | 50.20 | 52.86 | 52.31 | 51.70 | 33.38 | 31.65 | 33.29 | 30.45 | 32.19 |
| CSDG | 66.91 | 68.06 | 64.43 | 52.24 | 62.91 | 37.10 | 51.76 | 70.64 | 41.38 | 50.22 |
| CIConv | 67.42 | 70.83 | 65.19 | 42.77 | 61.55 | 45.40 | 45.38 | 57.08 | 32.44 | 45.08 |
| CIConv* | 78.38 | 75.67 | 69.33 | 55.92 | 69.83 | 45.75 | 50.64 | 71.93 | 35.33 | 50.91 |
| EGSDG w/o Aug. | 73.67 | 72.45 | 57.31 | 57.42 | 65.21 | 54.11 | 53.41 | 62.74 | 32.86 | 50.78 |
| EGSDG w/ Aug. | 73.45 | 78.48 | 71.94 | 60.13 | 71.00 | 55.14 | 57.34 | 72.50 | 45.84 | 57.71 |

## 4. Experiments and Results

### 4.1. Experimental Setup

**Datasets and Preprocessing** In our experiments, we employ three datasets, the cross-sequence brain tumor segmentation dataset (BraTS'19, T2 as source domain) (Menze et al., 2015), the cross-center prostate dataset (Prostate, Site B as source domain) (Liu et al., 2020), and the cross-modality cardiac dataset (MMWHS, CT and MRI as source domain respectively) (Zhuang and Shen, 2016), for evaluation. In particular, we shuffle all the volumes and divide them into four equal parts for each sequence firstly to prevent the ground truth leakage because the mask of each case is shared with four sequences in BraTS'19. Details are given in Appendix E.

**Network and Training Details** Following CSDG (Ouyang et al., 2021), we utilize U-Net (Ronneberger et al., 2015) with an EffcientNet-b2 (Tan and Le, 2019) backbone as our segmentation model and implement our model by PyTorch framework on one NVIDIA TITAN XP GPU. We use Adam optimizer (Kingma and Ba, 2014) with an initial learning rate of $3 \times 10^{-4}$ and batch size of 8 to train the model. For all experiments, the learning rate is decayed according to the polynomial rule for stable training.

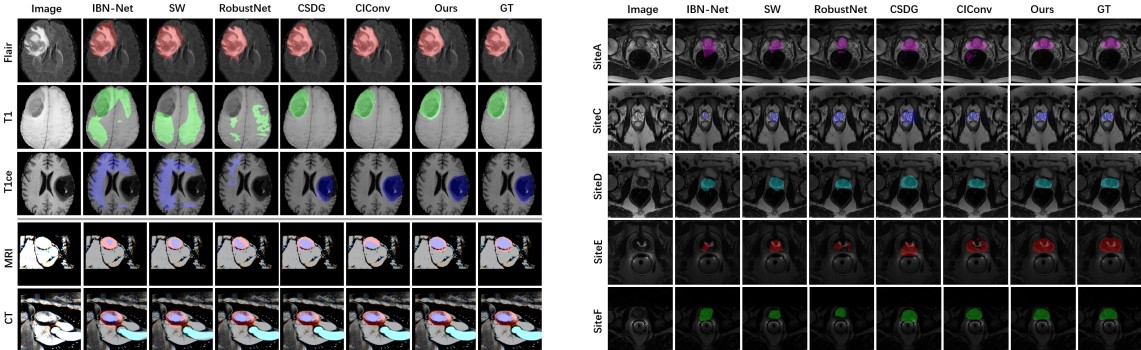

Figure 2: Qualitative comparison of BraTS'19 (top) and MMWHS (bottom) samples segmentation (left) and Prostate samples segmentation (right). MRI means CT→MRI domain generalization and CT means MRI→CT domain generalization.

**Evaluation Metrics** We take the Dice coefficient (Dice) as our evaluation metric, which measures the overlapping ratio between prediction and ground truth. The higher the Dice value, the better the segmentation performance.

### 4.2. Comparison Experiments

We compare our method with SOTA single-source domain generalization methods including IBN-Net (Pan et al., 2018), SW (Pan et al., 2019), RobustNet (Choi et al., 2021), SADN (Zhou et al., 2022), CSDG (Ouyang et al., 2021), and CIConv (Lengyel et al., 2021). For a fair comparison, we employ the same segmentation network to train the CIConv model. Besides, we also provide the results without domain generalization by directly applying the model learned in the source domain to unknown target domains (Lower Bound) and with supervised training on the target domain (Upper Bound). In addition, the comprehensive comparison between CIConv and ours is given in Appendix F.

Table 1 and Table 2 report the comparison results on the BraTS'19, Prostate, and MMWHS datasets respectively. Overall, our model outperforms others, especially in the large distribution shift dataset (BraTS'19 and MMWHS). For the results of normalization and whitening-based models (IBN-Net, SW, and RobustNet), which are designed for nature image, their performance is significantly lower than our model in each evaluation dataset. For the results of data augmentation-based methods (SADN and CSDG), their performance is unstable on different datasets. The distribution of grayscale values varies across different datasets and the level of domain shift varies among different SDG segmentation settings. However, the augmented samples fail to cover the unseen target domain distribution on the specific task and lead to terrible generalization performance. In addition, the performance of CIConv is lower than our model on all three datasets. Qualitative examples are shown in Figure 2. As we can see, our model can produce accurate and sharp predictions at the boundary of anatomies. The enlarged qualitative results refer to Appendix G.

Table 3: Performance of different edge-guided strategies on BraTS'19 (left) and Prostate (right) with vanilla U-Net.

| Experiments | Source Domain: T2 | | | |
|---|---|---|---|---|
| | T1 | T1ce | Flair | Avg. |
| Lower Bound | 10.72 | 5.86 | 58.34 | 24.97 |
| Exp. 1 (Ours) | 37.86 | 39.95 | 51.10 | **42.97** |
| Exp. 2 | 11.65 | 11.42 | 61.02 | 28.03 |
| Exp. 3 | 13.90 | 11.37 | 58.88 | 28.05 |
| Exp. 4 | 11.58 | 9.22 | 59.48 | 26.76 |
| Exp. 5 | 10.14 | 9.35 | 64.84 | 28.11 |
| Exp. 6 | 13.28 | 10.57 | 58.93 | 27.59 |
| Exp. 7 | 10.60 | 8.92 | 64.48 | 28.00 |

| Experiments | Source Domain: Site B | | | | | |
|---|---|---|---|---|---|---|
| | Site A | Site C | Site D | Site E | Site F | Avg. |
| Lower Bound | 42.25 | 25.79 | 59.91 | 14.88 | 37.12 | 35.99 |
| Exp. 1 (Ours) | 47.79 | 31.39 | 48.68 | 52.01 | 51.34 | **46.24** |
| Exp. 2 | 58.65 | 42.94 | 46.88 | 12.81 | 31.87 | 38.63 |
| Exp. 3 | 38.31 | 17.44 | 61.78 | 21.68 | 41.35 | 36.11 |
| Exp. 4 | 32.33 | 24.74 | 43.97 | 32.14 | 36.14 | 33.86 |
| Exp. 5 | 48.3 | 37.14 | 66.96 | 20.48 | 35.02 | 41.58 |
| Exp. 6 | 43.56 | 25.73 | 78.45 | 20.3 | 55.23 | 44.65 |
| Exp. 7 | 51.87 | 27.15 | 61.72 | 18.16 | 49.23 | 41.63 |

Table 4: Performance of different edge-guided strategies on MMWHS with vanilla U-Net.

| Experiments | Source Domain: MRI | | | | | Source Domain: CT | | | | |
|---|---|---|---|---|---|---|---|---|---|---|
| | AA | LAC | LVC | MYO | Avg. | AA | LAC | LVC | MYO | Avg. |
| Lower Bound | 22.60 | 36.71 | 23.51 | 13.08 | 23.98 | 12.84 | 37.94 | 23.44 | 5.45 | 19.92 |
| Exp. 1 (Ours) | 63.37 | 64.05 | 39.76 | 38.50 | **51.42** | 43.74 | 52.79 | 60.39 | 34.59 | **47.88** |
| Exp. 2 | 53.95 | 51.45 | 42.73 | 31.10 | 44.81 | 23.78 | 38.40 | 36.22 | 14.02 | 28.10 |
| Exp. 3 | 49.67 | 54.07 | 39.19 | 25.58 | 42.13 | 21.70 | 38.73 | 43.90 | 13.28 | 29.40 |
| Exp. 4 | 50.63 | 53.15 | 45.02 | 25.90 | 43.68 | 29.92 | 41.07 | 39.01 | 12.59 | 30.65 |
| Exp. 5 | 50.26 | 52.04 | 44.76 | 29.86 | 44.23 | 30.11 | 48.25 | 41.46 | 13.62 | 33.36 |
| Exp. 6 | 50.14 | 49.72 | 34.54 | 21.74 | 39.03 | 25.52 | 38.94 | 44.98 | 11.54 | 30.24 |
| Exp. 7 | 57.50 | 52.66 | 36.08 | 26.36 | 43.15 | 19.80 | 46.68 | 25.99 | 10.41 | 25.72 |

### 4.3. Edge-guided Strategy

To enhance the model's attention at the boundary of targets, previous works (Lengyel et al., 2021; Liu et al., 2021, 2022; You et al., 2023) have tried different strategies. Here, we conducted a comprehensive analysis to the impact of image edge on the model generalization ability. We employ the vanilla U-Net (Ronneberger et al., 2015) as the segmentation network which is borrowed from Pytorch-UNet. Other configurations are the same as the main experiment. The visualization of different edge-guided strategies and corresponding explanations are provided in Figure 3. These edge-guided strategies can be divided into two categories: (i) using the edge map as the supervision signal to increase the model's attention at the boundary of segmentation targets. (ii) concatenating the feature map and edge map to force the model learning domain invariant representation and enhance the generalization ability. Note that we make the same process at the testing stage for the second category experiments (Exp. 1, 2, 3, and 4).

Table 3 and Table 4 report the comparison results on BraTS'19, Prostate, and MMWHS datasets respectively. Overall, adopting the edge map as an additional guided signal can improve model generalization performance compared to the Lower Bound. Notably, employing the image edge as input to train the model directly brings tremendous generalization ability gains on the three datasets. We note that (i) for the former, the performance is lower than ours, probably because this category strategy weakens the supervision of the edge information via the segmentation head at the training stage, and (ii) for the latter, the performance is lower than ours, possibly due to the grayscale information making the model overfit on the source domain.

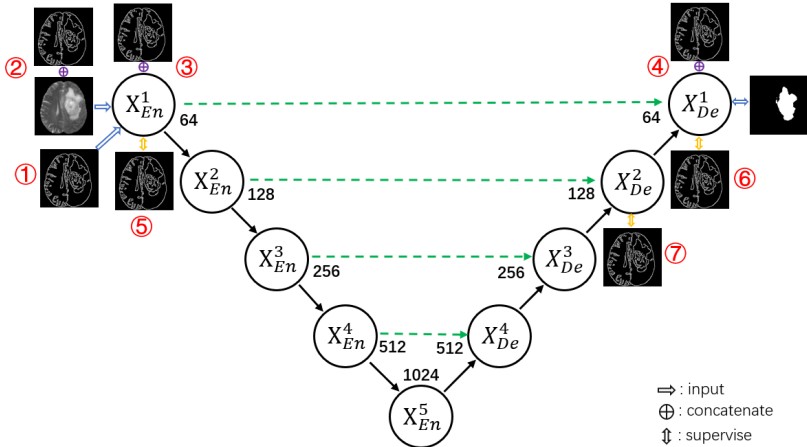

Figure 3: Visualization of different edge-guided strategies with vanilla U-Net framework. Exp. 1 means taking the image edge as input. Exp. 2 means concatenating the image and edge map as input. Exp. 3 means concatenating the feature map $X_{En}^1$ and the edge map as the next layer's input. Exp. 4 means concatenating the feature map $X_{De}^1$ and the edge map as the segmentation layer's input. Exp. 5 means employing the image edge as the supervision of feature map $X_{En}^1$ with a single Conv2d segmentation layer. Exp. 6 means employing the image edge as the supervision of feature map $X_{De}^1$ with a single Conv2d segmentation layer. Exp. 7 means employing the image edge as the supervision of feature map $X_{De}^2$ with a single Conv2d segmentation layer. AutoCanny is employed in all experiments.

## 5. Conclusion and Discussion

In this work, we use the edge of image as input to train a network. It improves the model's generalization performance significantly on unknown target domains. Extensive experiments on three typical medical image segmentation datasets demonstrate our approach achieves superior generalization performance compared to the state-of-the-art SDG methods. However, our model has the following limitations: (i) The optimal edge extractor is different in diverse segmentation scenarios, which brings great challenges to choosing the best one for an unseen dataset. (ii) In low-contrast images (like Ultrasound or CT), the model cannot extract valuable edge information well, which may lead to poor segmentation performance. (iii) When the segmentation target is small (like multiple sclerosis or cochlear), the extracted edge information may be similar to the surrounding noise, which will lead to the wrong segmentation results. In addition, there are limitations in the experimental setup of BraTS'19 because different sequences focus on different structures, which may lead to the tumor is not well visible in on specific modality.

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

## Appendix A. Edge detector operators

Digital images are discrete and different edge detection algorithms differ in the way of $G_x$ and $G_y$ calculation. Different edge detector operators are given in Figure 4.

| Edge Detector | Definition | Operator |
|---|---|---|
| Roberts | $G_x = f(x,y) - f(x+1,y+1)$
$G_y = f(x,y+1) - f(x+1,y)$ | $G_x = \begin{bmatrix} 1 & 0 \\ 0 & -1 \end{bmatrix} \ G_y = \begin{bmatrix} 0 & 1 \\ -1 & 0 \end{bmatrix}$ |
| Sobel | $G_x = \{f(x+1,y-1) + 2f(x+1,y)$
$\quad + f(x+1,y+1)\} -$
$\{f(x-1,y-1) + 2f(x-1,y)$
$\quad + f(x-1,y+1)\}$
$G_y = \{f(x-1,y+1) + 2f(x,y+1)$
$\quad + f(x+1,y+1)\} -$
$\{f(x-1,y-1) + 2f(x,y-1)$
$\quad + f(x+1,y-1)\}$ | $G_x = \begin{bmatrix} -1 & -2 & -1 \\ 0 & 0 & 0 \\ 1 & 2 & 1 \end{bmatrix}$

$G_y = \begin{bmatrix} -1 & 0 & 1 \\ -2 & 0 & 2 \\ -1 & 0 & 1 \end{bmatrix}$ |
| Prewitt | $G_x = \{f(x+1,y-1) + f(x+1,y)$
$\quad + f(x+1,y+1)\} -$
$\{f(x-1,y-1) + f(x-1,y)$
$\quad + f(x-1,y+1)\}$
$G_y = \{f(x-1,y+1) + f(x,y+1)$
$\quad + f(x+1,y+1)\} -$
$\{f(x-1,y-1) + f(x,y-1)$
$\quad + f(x+1,y-1)\}$ | $G_x = \begin{bmatrix} -1 & -1 & -1 \\ 0 & 0 & 0 \\ 1 & 1 & 1 \end{bmatrix}$

$G_y = \begin{bmatrix} -1 & 0 & 1 \\ -1 & 0 & 1 \\ -1 & 0 & 1 \end{bmatrix}$ |
| Laplacian | $\nabla^2 f(x,y) = f(x+1,y) + f(x-1,y)$
$+ f(x,y+1) +$
$f(x,y-1) - 4f(x,y)$ | $\nabla^2 f(x,y) = \begin{bmatrix} 0 & 1 & 0 \\ 1 & -4 & 1 \\ 0 & 1 & 0 \end{bmatrix}$ |

Figure 4: Overview of classic edge detection algorithms' definitions and corresponding operators of the partial derivative of the image in the X and Y directions.

## Appendix B. Details and configuration of Canny algorithm

### B.1. Detail of Canny algorithm

The steps of the Canny algorithm (Canny, 1986) include Image Smoothing, Gradient Calculation, Non-maximum Suppression, and Edges Checking.

**Image Smoothing** Gaussian filter is used to smooth images and get rid of the noise which is defined as :

$$G(x,y) = \frac{1}{2\pi\sigma^2} exp(-\frac{x^2 + y^2}{2\sigma^2}) \tag{6}$$

where $\sigma$ stands for the size of the Gaussian Kernel, which controls the extent of smoothing the image. This critical parameter needs to be set manually based on experience.

**Gradient Calculation** The traditional Canny algorithm adopts a limited difference of $2 \times 2$ neighboring area to calculate the magnitude and direction of the image gradient. The operator of the partial derivative of the image in the X and Y directions is defined by:

$$G_x = \begin{bmatrix} -1 & 1 \\ -1 & 1 \end{bmatrix}, G_y = \begin{bmatrix} 1 & 1 \\ -1 & -1 \end{bmatrix} \tag{7}$$

**Non-maximum Suppression** After acquiring the gradient magnitude image, it's needed to perform non-maximum suppression (NMS) on the image to accurately locate edges. The process of NMS can help guarantee that each edge is one-pixel width.

**Edges Checking** Canny adopts a double-threshold method to select edge points after carrying on non-maximum suppression. The pixels whose gradient magnitude is above the high threshold will be marked as edge points, and those whose gradient magnitude is under the low threshold will be marked as non-edge points, and the rest will be marked as candidate edge points. Those candidate edge points that are connected with edge points will be marked as edge points. This method reduces the influence of noise on the edge of the final edge image. The low and high thresholds need to be set manually based on experience.

### B.2. Configuration of Canny algorithm

We have manually designed the corresponding Cany detector parameters for each domain of each dataset. For all scenarios, the size of the Gaussian kernel $3 \times 3$. The configuration of the double-threshold is provided in Table 5.

Table 5: The configuration of Canny algorithm's double-threshold.

| Dataset | Domain | Low | High |
|---|---|---|---|
| BraTS'19 | T2 | 40 | 80 |
| | T1 | 20 | 60 |
| | Flair | 40 | 100 |
| | T1ce | 20 | 50 |
| Prostate | Site A | 50 | 200 |
| | Site B | 100 | 200 |
| | Site C | 50 | 150 |
| | Site D | 50 | 140 |
| | Site E | 20 | 40 |
| | Site F | 30 | 70 |
| MMWHS | MRI | 30 | 80 |
| | CT | 70 | 120 |

## Appendix C. Visualization of edge detectors and CIConv

The visual comparison of different classic edge detection algorithms (Canny, 1986; Rong et al., 2014; Roberts, 1963; Prewitt et al., 1970; Kittler, 1983; LeCun et al., 1998) and CIConv (Lengyel et al., 2021) is shown in Figure 5. As we can see, there are large differences in the edge or gradient map extracted by different edge detection algorithms with the same image. Compared with CIConv, the classic edge detection algorithms can filter more useless information with less computation.

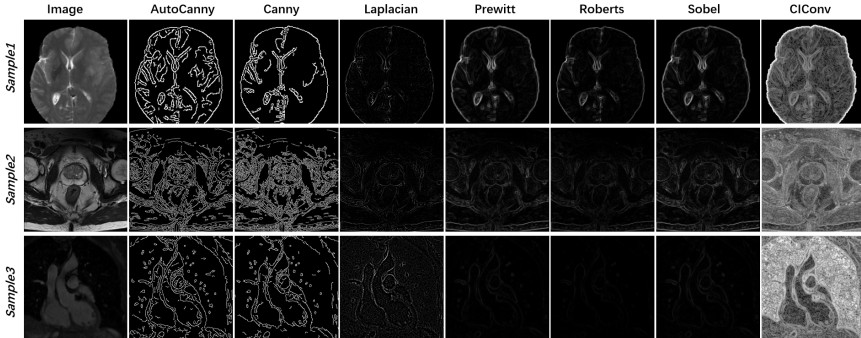

Figure 5: Visualization comparison of different edge detection algorithms and CIConv.

## Appendix D. Edge-guided Model with Data Augmentation

### D.1. Visualization of BézierCurve augmentation

The visualization examples of BézierCurve augmentation are shown in Figure 6. As introduced in the main text, this augmentation method maps the source image to diverse grayscale value distribution and keeps the appearance of the anatomic structures perceivable at the same time.

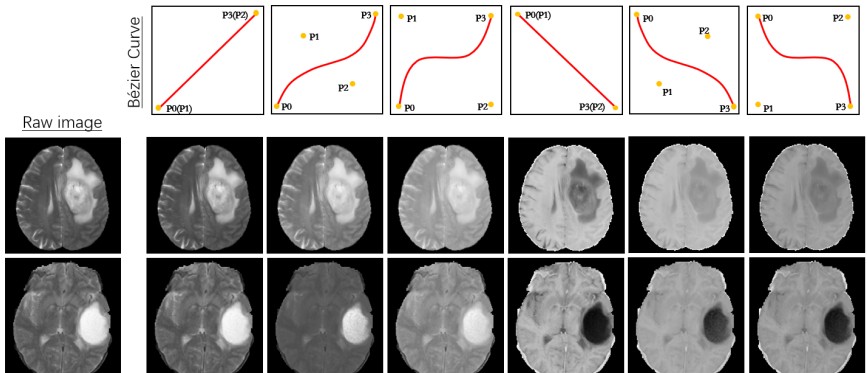

Figure 6: Visualization of generated Bezier Curve and corresponding augmented image on the BraTS'19 samples.

### D.2. Edge and gradient map of BézierCurve augmented image

We realized that for the same case, the edge or gradient map in each domain is different by the same edge detector. Therefore, we use data augmentation to simulate the data distribution of unknown target domain before edge extraction to enrich the gradient information of edge-guided training. The visualization of the BézierCurve augmented image and corresponding edge and gradient map is shown in Figure 7.

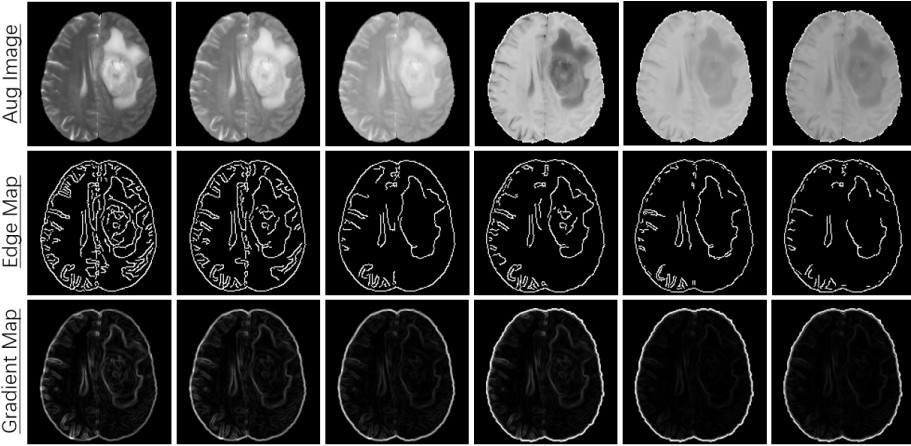

Figure 7: Visualization of augmented image and corresponding edge (AutoCanny) and gradient (Sobel) map.

### D.3. Results of Edge-guided models with BézierCurve

**Edge Detector** Different edge detectors will extract distinct image edges or gradients for the same image, which affects the training process and testing performance. Accordingly, we conducted comprehensive comparison experiments on classical edge detectors, including Canny (Canny, 1986), AutoCanny (Rong et al., 2014), Roberts (Roberts, 1963), Prewitt (Prewitt et al., 1970), Sobel (Kittler, 1983), and Laplacian (LeCun et al., 1998).
**BézierCurve Augmentation** For edge-guided models, data augmentation is supposed to simulate the edge or gradient information of the unseen target samples to train a model with great generalization ability. To this end, we also explore the above edge-guided models on the BézierCurve (Zhou et al., 2019) augmented samples, which is a simple idea to generate different styles by adjusting the gray value distribution of images.

Table 6 and Table 7 show the results. As we can see, taking the edge as input promotes the generalization ability of the model remarkably and the BézierCurve can further improve its performance. We note that different edge extractors are sensitive to specific types of edges like edge orientation, noise environment, and edge structure. In different segmentation scenarios, the texture, intensity, and noise of medical images are diverse. This leads to the discrepancy between the valuable edge extraction and the irrelevant noise filtering by different edge extractors, which makes the optimal edge detector on each domain different. This will bring great challenges to choosing the best one for an unseen dataset.

Table 6: Ablation study of edge detectors with original (first group) and BézierCurve augmented (second group†) sample on BraTS'19 (left) and Prostate (right).

| Edge Detector | Source Domain: T2 | | | | Edge Detector | Source Domain: Site B | | | | | |
|---|---|---|---|---|---|---|---|---|---|---|---|
| | T1 | T1ce | Flair | Avg. | | Site A | Site C | Site D | Site E | Site F | Avg. |
| Canny | 40.30 | 50.18 | 61.96 | 50.81 | Canny | 72.12 | 46.13 | 64.82 | 63.93 | 62.74 | 61.95 |
| AutoCanny | 48.67 | 56.25 | 66.07 | 57.0 | AutoCanny | 72.70 | 59.54 | 83.00 | 70.36 | 81.11 | **73.34** |
| Roberts | 43.70 | 48.93 | 67.40 | 53.34 | Roberts | 68.04 | 49.73 | 75.93 | 71.82 | 78.79 | 68.86 |
| Prewitt | 50.28 | 50.31 | 72.38 | 57.66 | Prewitt | 73.39 | 48.83 | 81.08 | 80.27 | 69.05 | 70.52 |
| Sobel | 51.38 | 50.35 | 71.63 | **57.79** | Sobel | 73.35 | 48.36 | 84.13 | 79.95 | 71.07 | 71.37 |
| Laplacian | 31.39 | 43.39 | 61.86 | 45.55 | Laplacian | 73.48 | 50.19 | 81.20 | 79.92 | 81.74 | 73.31 |
| Canny† | 56.58 | 53.48 | 62.39 | 57.48 | Canny† | 66.28 | 56.55 | 58.59 | 70.29 | 66.43 | 63.63 |
| AutoCanny† | 56.28 | 53.76 | 63.84 | 57.96 | AutoCanny† | 78.51 | 64.16 | 82.95 | 77.34 | 78.20 | **76.23** |
| Roberts† | 58.66 | 55.60 | 68.84 | 61.03 | Roberts† | 75.23 | 57.62 | 80.43 | 79.45 | 71.26 | 72.80 |
| Prewitt† | 55.26 | 55.82 | 72.59 | 61.22 | Prewitt† | 71.40 | 56.59 | 76.44 | 79.50 | 77.95 | 72.38 |
| Sobel† | 62.59 | 54.68 | 77.07 | **64.78** | Sobel† | 75.48 | 55.87 | 75.53 | 84.10 | 70.60 | 72.32 |
| Laplacian† | 54.55 | 55.78 | 58.68 | 56.34 | Laplacian† | 75.66 | 42.04 | 85.28 | 83.30 | 82.89 | 73.83 |

Table 7: Ablation study of edge detectors with original (top) and BézierCurve augmented (bottom†) sample on MMWHS.

| Edge Detector | Source Domain: MRI | | | | | Source Domain: CT | | | | |
|---|---|---|---|---|---|---|---|---|---|---|
| | AA | LAC | LVC | MYO | Avg. | AA | LAC | LVC | MYO | Avg. |
| Canny | 67.91 | 69.87 | 67.63 | 50.15 | 63.89 | 54.11 | 53.41 | 62.74 | 32.86 | **50.78** |
| AutoCanny | 65.22 | 71.51 | 64.22 | 51.58 | 63.13 | 44.96 | 56.68 | 58.79 | 34.93 | 48.84 |
| Roberts | 72.17 | 70.72 | 59.21 | 55.41 | 64.38 | 32.21 | 49.15 | 52.98 | 18.96 | 38.33 |
| Prewitt | 68.76 | 70.82 | 65.87 | 51.67 | 64.28 | 37.98 | 49.31 | 61.04 | 21.92 | 42.56 |
| Sobel | 73.67 | 72.45 | 57.31 | 57.42 | **65.21** | 40.80 | 54.55 | 64.92 | 21.94 | 45.55 |
| Laplacian | 67.43 | 72.02 | 62.72 | 56.52 | 64.67 | 36.27 | 48.94 | 74.07 | 30.79 | 47.52 |
| Canny† | 66.03 | 73.65 | 71.11 | 52.69 | 65.87 | 55.14 | 57.34 | 72.50 | 45.84 | **57.71** |
| AutoCanny† | 70.63 | 69.81 | 67.15 | 52.39 | 65.00 | 52.89 | 62.96 | 65.15 | 34.48 | 53.87 |
| Roberts† | 72.04 | 73.90 | 65.43 | 54.89 | 66.57 | 46.38 | 46.66 | 65.67 | 29.47 | 47.04 |
| Prewitt† | 71.05 | 75.04 | 68.31 | 55.07 | 67.37 | 49.40 | 54.18 | 60.27 | 32.12 | 48.99 |
| Sobel† | 73.45 | 78.48 | 71.94 | 60.13 | **71.00** | 35.57 | 52.98 | 60.45 | 31.96 | 45.24 |
| Laplacian† | 70.74 | 70.23 | 64.51 | 54.79 | 65.07 | 49.99 | 49.18 | 67.71 | 32.65 | 49.88 |

## D.4. Results of Edge-guided models with RandConv

To further validate the effectiveness of data augmentation to edge-guided models. We conducted an experiment on Edge-guided with RandConv (Xu et al., 2020), which employs transformation via randomly initializing the weight of the first convolution layer. Table 8 reports the results on three datasets, which shows that the performance has improved compared to using only the edge-guided model. However, it's generally lower than training edge-guided models with BézierCurve augmented samples.

Table 8: The result of edge-guided methods with RandConv augmented sample on the BraTS'19 (left), Prostate (middle), and MMWHS (right) datasets.

| Edge Detector | Source Domain: T2 | | | | Edge Detector | Source Domain: Site B | | | | | | Edge Detector | Source Domain: MRI | | | | |
|---|---|---|---|---|---|---|---|---|---|---|---|---|---|---|---|---|---|
| | T1 | T1ce | Flair | Avg. | | Site A | Site C | Site D | Site E | Site F | Avg. | | AA | LAC | LVC | MYO | Avg. |
| Canny | 53.48 | 53.00 | 57.80 | 54.76 | Canny | 57.30 | 52.65 | 58.57 | 58.73 | 45.65 | 54.58 | Canny | 63.99 | 70.52 | 58.85 | 52.10 | 61.37 |
| AutoCanny | 38.85 | 40.13 | 65.22 | 48.07 | AutoCanny | 76.27 | 59.08 | 80.09 | 74.93 | 72.56 | **72.59** | AutoCanny | 68.18 | 68.96 | 59.79 | 48.04 | 61.24 |
| Roberts | 51.43 | 46.49 | 67.82 | 55.25 | Roberts | 70.75 | 50.53 | 76.69 | 52.93 | 65.61 | 63.30 | Roberts | 69.44 | 69.65 | 57.90 | 47.74 | 61.18 |
| Prewitt | 46.81 | 51.46 | 68.37 | 55.55 | Prewitt | 62.70 | 52.44 | 54.53 | 54.46 | 46.41 | 54.11 | Prewitt | 65.77 | 65.04 | 49.12 | 55.01 | 58.74 |
| Sobel | 56.32 | 51.90 | 67.68 | **58.63** | Sobel | 73.72 | 53.78 | 75.41 | 64.95 | 54.90 | 64.55 | Sobel | 72.58 | 69.04 | 66.96 | 57.96 | **66.64** |
| Laplacian | 51.30 | 53.67 | 68.32 | 57.76 | Laplacian | 65.99 | 60.79 | 61.20 | 56.81 | 32.22 | 55.40 | Laplacian | 75.93 | 68.83 | 65.15 | 55.59 | 66.38 |

## Appendix E. Details of datasets and preprocessing

BraTS'19 contains 335 cases which were acquired with different clinical protocols and various scanners from multiple institutions. Each case is composed of four sequences of MR images (T2, T1, Flair, and T1CE). Due to experts always annotating the whole tumor on T2, we use T2 as the source domain and others as unknown target domains. Prostate contains prostate T2-weighted MRI data collected from six different data sources. We follow the previous work (Liu et al., 2020) to partition the data into six datasets A to F, according to the clinical centers that the datasets collected. Consistent with our previous work, we take Site B as the source domain and others as unknown target domains. MMWHS dataset consists of unpaired 20 MRI and 20 CT volumes collected at different clinical sites, which contains the ground truth mask of four cardiac structures, including the ascending aorta (AA), the left atrium blood cavity (LAC), the left ventricle blood cavity (LVC), and the myocardium of the left ventricle (MYO). We make domain generalizations in both directions.

For data preprocessing, each volume was normalized to zero mean and unit variance. Then, we get the slices from each volume in the axial (BraTS'19 and Prostate) or coronal (MMWHS) plane and normalize the image to [-1, 1] before feeding it to the network. For BraTS'19 and MMWHS, we make the center crop and then resize it to $256 \times 256$. For Prostate, the size of the image is $384 \times 384$. Each domain was randomly split with 80% samples for training and 20% samples for testing. It is worth noting that (i) there are three sub-structures in BraTS'19 (the Enhancing Tumor (ET), the Tumor Core (TC), and the Whole Tumor (WT)) and we merged them into one label for segmentation which is consistent with SADN (Zhou et al., 2022) and (ii) we shuffle all the volumes and divide them into four equal parts for each sequence firstly to prevent the ground truth leakage problem because the mask of each case is shared with four sequences in BraTS'19.

## Appendix F. Comparison between ours and CIConv

In essence, Color Invariant Convolution (CIConv) (Lengyel et al., 2021) is a learnable edge detector that is derived from the physics-based reflection models (Geusebroek et al., 2001). On the one hand, the computational process of Color Invariant theory (Geusebroek et al., 2001) is very complicated, which greatly increases the training and inference time. On the other hand, the performance of different variants of CIConv is unstable. Therefore, we compare the performance of CIConv and edge detection algorithms with the RefineNet (Lin et al., 2017) which is utilized in the CIConv model. Table 9 reports the comparison results, where we can see that the performance of different variants in CIConv varies greatly, while the performance of all edge detectors is stable and superior to the CIConv.

Table 9: The comparison result of CIConv (first group) and Edge-guided model (second group) on the BraTS'19 (left), Prostate (middle), and MMWHS (right) datasets with RefineNet. The best performance of CIConv and Edge-guided model is underlined and bolded respectively.

| Experiment | Source Domain: T2 | | | | Experiment | Source Domain: Site B | | | | | | Experiment | Source Domain: MRI | | | | |
|---|---|---|---|---|---|---|---|---|---|---|---|---|---|---|---|---|---|
| | T1 | T1ce | Flair | Avg. | | Site A | Site C | Site D | Site E | Site F | Avg. | | AA | LAC | LVC | MYO | Avg. |
| invariant-E | 34.95 | 41.08 | 62.06 | 46.03 | invariant-E | 58.61 | 55.81 | 68.07 | 46.46 | 58.25 | 57.44 | invariant-E | 55.55 | 57.07 | 52.77 | 26.79 | 48.04 |
| invariant-W | 44.79 | 31.15 | 57.10 | 44.35 | invariant-W | 57.71 | 42.28 | 70.78 | 37.61 | 56.84 | 53.04 | invariant-W | 70.35 | 76.32 | 58.81 | 42.45 | 61.98 |
| invariant-C | 10.93 | 18.98 | 25.00 | 18.30 | invariant-C | 52.59 | 40.85 | 68.56 | 63.49 | 64.06 | 57.91 | invariant-C | 60.17 | 60.09 | 45.79 | 30.00 | 49.01 |
| invariant-N | 0.00 | 0.00 | 0.00 | 0.00 | invariant-N | 0.00 | 0.00 | 0.00 | 0.00 | 0.00 | 0.00 | invariant-N | 0.00 | 0.00 | 0.00 | 0.00 | 0.00 |
| invariant-H | 7.98 | 10.78 | 10.80 | 9.85 | invariant-H | 55.06 | 45.26 | 56.62 | 57.28 | 63.96 | 55.64 | invariant-H | 60.52 | 63.39 | 53.77 | 31.59 | 52.32 |
| Canny | 42.82 | 45.62 | 51.34 | 46.59 | Canny | 60.02 | 52.19 | 63.12 | 74.62 | 64.56 | 62.9 | Canny | 70.10 | 73.30 | 63.79 | 43.81 | 62.75 |
| AutoCanny | 39.87 | 45.45 | 51.69 | 45.67 | AutoCanny | 77.10 | 62.35 | 71.05 | 71.00 | 66.54 | 69.61 | AutoCanny | 67.56 | 71.43 | 54.27 | 44.45 | 59.43 |
| Roberts | 38.21 | 42.95 | 51.71 | 44.29 | Roberts | 63.31 | 40.56 | 66.35 | 39.74 | 42.55 | 50.50 | Roberts | 71.93 | 70.81 | 64.53 | 54.83 | 65.53 |
| Prewitt | 36.62 | 36.73 | 60.55 | 44.63 | Prewitt | 62.25 | 47.78 | 71.78 | 60.48 | 64.69 | 61.40 | Prewitt | 66.75 | 72.95 | 65.67 | 54.29 | 64.92 |
| Sobel | 38.41 | 44.14 | 59.36 | 47.30 | Sobel | 61.25 | 36.48 | 69.08 | 51.35 | 42.87 | 52.21 | Sobel | 64.85 | 75.37 | 66.26 | 56.15 | 65.66 |
| Laplacian | 19.60 | 22.54 | 49.32 | 30.49 | Laplacian | 64.70 | 36.69 | 69.26 | 53.29 | 51.85 | 55.16 | Laplacian | 71.99 | 72.48 | 64.33 | 51.75 | 65.14 |

## Appendix G. The enlarged qualitative results

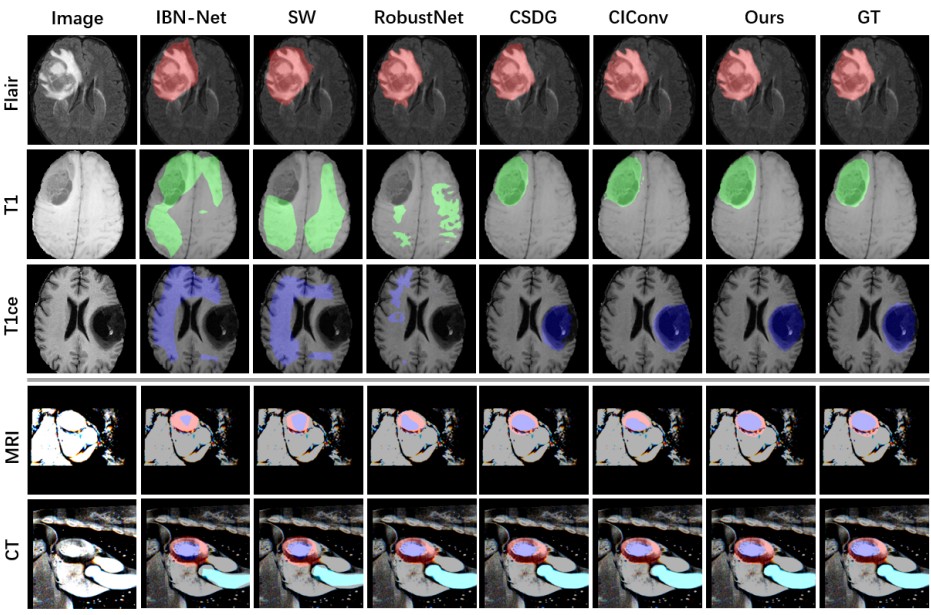

Figure 8: Qualitative comparison of BraTS'19 (top) and MMWHS (bottom) samples. MRI means CT→MRI domain generalization and CT means MRI→CT domain generalization.

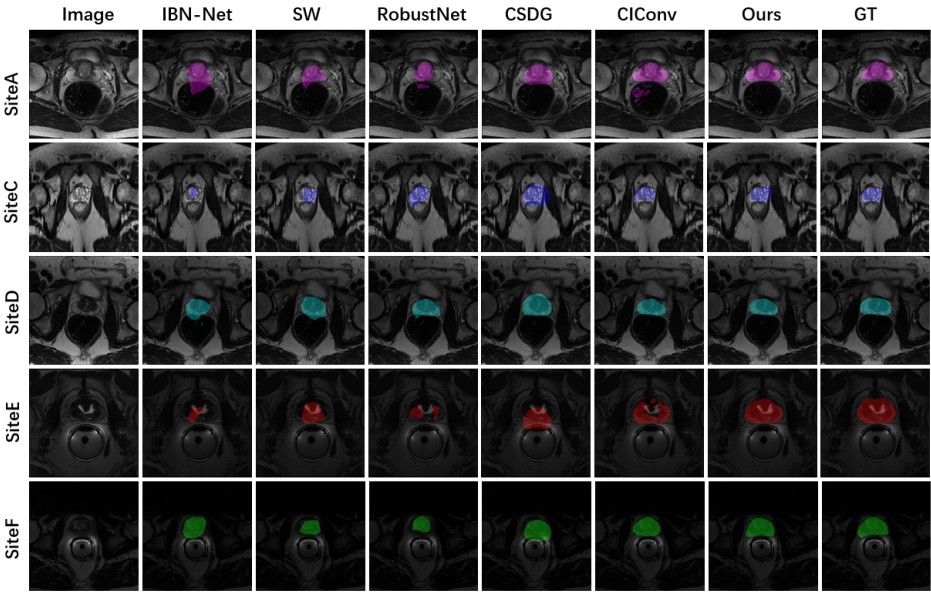

Figure 9: Qualitative comparison of Prostate samples.

