# OpenReview forum: "Train Once, Deploy Anywhere: Edge-Guided Single-source Domain Generalization for Medical Image Segmentation"
_MIDL.io/2024/Conference — MIDL 2024 Poster_

### Official Review · Reviewer_CChP · 2024-02-27

**Confidence:** 4
**Preliminary Rating:** 2
**Final Rating:** 4

**Summary:**

The authors propose a single-source domain generalization method for medical image segmentation.

Surprisingly simple, it applies edge detection and augmentations to the single source domain and trains a regular segmentation network on the masks.

**Strengths:**

x. Leveraging edge detection and corresponding data augmentation method (BezierCurve) from classical image processing techniques to achieve single-source domain generalization is **inspiring**.

x. The results on the prostate dataset look good.

**Weaknesses:**

Although the presented methodology is interesting in the context of single-source DG, there are a few key concerns:

x. **Experimental setting for brain tumor segmentation is incorrect**.

Generalizing T2 to other modalities is not a good setting because different tumor structures are annotated from different modalities. For example, the segmentation labels one could see from FLAIR and T2 are different.

x. **Limited comparison on public datasets.**

While one of the datasets used for evaluation is not well curated, the authors didn't compare their method with others on other 'standard' datasets like, abdominal CT-MRI, and Cardiac bSSFP-LGE in Ouyang et al's work (the CSDG method).

More comparisons would probably highlight the advantages and disadvantages of the proposed method. It would be interesting to discuss about it.

x. **Limited discussion about the proposed method.**

While edge detection shows promising results, some discussions are needed regarding the limitations.

For example, it might be degraded when the texture information (instead of shape or edge) is dominant or when the contour is not clearly visible, for example, ultrasound or CT images.

x. **The title is very confusing.**

'Adapt' in the title might give the reader an impression that they are doing 'domain adaptation'. In reality, the authors are just doing domain generalization.

I suggest the authors change the title, like just 'Edge-Guided Single-source Domain Generalization for Medical Image Segmentation'.

**Detailed Comments:**

My detailed constructive comments are the same as above.

I **encourage** the authors to thoroughly discuss the advantages and disadvantages of this work by showing results on diverse datasets (also for more fair comparisons.)

**Justification Of Final Rating:**

I thank the communication from the authors.
The authors are responsive and address most of my concerns, including the confusion about the original title, an issue in the BraTS experimental setup, and the limitations of this work.

I raised my score from borderline to borderline accept, and then to weak accept.

**Justification Of The Preliminary Rating:**

The paper has a clear fault in one of the experiments (brain tumor segmentation).

It has limited comparisons on available domain generalization datasets.

It needs more discussion on the advantages and disadvantages of this work, although the general idea sounds inspiring.

**Questions To Address In The Rebuttal:**

Please see the weakness part.

The setting for the brain tumor is wrong.

Please check the CSDG paper for detailed datasets used for comparisons.

For the other points, please check the weakness part.

**Special Issue:**

No

---

> ### Author Response · Authors · 2024-03-14
>
> Thanks for your useful comments sincerely.
> - Yes, we were aware of this issue and made special preprocessing on the BraTS dataset, which has been explained in Appendix G. Concretely, we shuffle all the volumes and divide them into four equal parts for each sequence firstly to prevent the ground truth leakage because the mask of each case is shared with four sequences in BraTS. In the final version, we will place this important information on the main content.
>
> - We consider that the cross-modality and cross-sequence experiment settings have covered the above two situations. In addition, our experimental setup is consistent with existing SDG methods (BraTS[1], Prostate[2], MMWHS[1]).
> > [1] Zhou Z, Qi L, Yang X, et al. Generalizable cross-modality medical image segmentation via style augmentation and dual normalization[C]//Proceedings of the IEEE/CVF conference on computer vision and pattern recognition. 2022: 20856-20865.
> > [2] Ouyang C, Chen C, Li S, et al. Causality-inspired single-source domain generalization for medical image segmentation[J]. IEEE Transactions on Medical Imaging, 2022, 42(4): 1095-1106.
>
> - Yes, the model has the following limitations:
>     1. The optimal edge extractor is diverse in different segmentation scenarios, which brings great challenges to choosing the best one for an unseen dataset.
>     2. In low-contrast images (like Ultrasound or CT), the model cannot extract valuable edge information well, which may lead to poor segmentation performance.
>     3. When the segmentation target is small (like multiple sclerosis or cochlear), the extracted edge information may be similar to the surrounding noise, which may lead to the wrong segmentation of the target and noise.
>
> - Yes, we have the same consideration when choosing the title. We note that, in essence, both Domain Generalization and Domain Adaptation are doing the "Adapt" operation, so we choose "Train Once, Adapt Anywhere".
>
> - For more fair comparisons, the results of the Edge-guided model without BézierCurve augmentation are as follows:
> |Methods|T1|T1ce|Flair|Average|
> |-|-|-|-|-|
> |UpperBound|74.42|71.64|82.75|76.27|
> |LowerBound|13.82|11.58|66.61|30.67|
> |IBN-Net|34.37|48.27|42.33|41.66|
> |SW|31.83|40.48|34.95|35.75|
> |RobustNet|8.59|10.14|68.29|29.01|
> |SADN|49.36|38.09|75.87|54.44|
> |CSDG|46.76|44.99|60.20|50.65|
> |CIConv|15.36|20.83|*76.07*|37.42|
> |CIConv*|*53.82*|*52.69*|74.05|*60.19*|
> |EGSDG w/o Aug.|51.38|50.35|71.63|57.79|
> |EGSDG w/ Aug.|**62.59**|**54.68**|**77.07**|**64.78**|
>
> |Methods|SiteA|SiteC|SiteD|SiteF|SiteE|Average|
> |-|-|-|-|-|-|-|
> |UpperBound|89.13|89.96|89.31|87.76|89.34|89.10|
> |LowerBound|63.62|19.42|81.06|83.89|71.17|63.83|
> |IBN-Net|67.36|46.79|65.09|71.45|76.88|65.51|
> |SW|70.83|51.71|70.89|51.96|68.97|62.87|
> |Robust|73.27|55.04|77.41|54.79|70.21|66.14|
> |SADN|62.57|41.46|49.14|60.63|68.85|56.63|
> |CSDG|69.75|61.47|74.27|76.31|70.54|70.47|
> |CIConv|73.48|*63.51*|80.80|62.15|74.93|70.97|
> |CIConv*|*76.41*|59.74|76.63|**78.10**|77.17|*73.61*|
> |EGSDG w/o Aug.|72.7|59.54|**83.00**|70.36|**81.11**|73.34|
> |EGSDG w/ Aug.|**78.51**|**64.16**|*82.95*|*77.34*|*78.20*|**76.23**|
>
> |Methods|AA|LAC|LVC|MYO|Average|AA|LAC|LVC|MYO|Average|
> |-|-|-|-|-|-|-|-|-|-|-|
> |UpperBound|89.74|84.99|87.44|83.34|86.38|80.76|82.29|92.38|78.23|83.42|
> |LowerBound|32.18|35.92|19.53|9.42|24.26|18.44|8.84|38.72|9.65|18.91|
> |IBN-Net|59.04|67.63|67.34|45.49|59.88|31.23|42.36|59.91|34.63|42.03|
> |SW|52.94|69.52|64.28|44.64|57.84|38.95|47.62|62.82|33.30|45.67|
> |Robust|68.07|74.68|62.56|46.09|62.85|52.27|60.08|67.26|32.97|*53.14*|
> |SADN(released)|51.42|50.20|52.86|52.31|51.70|33.38|31.65|33.29|30.45|32.19|
> |CSDG|66.91|68.06|64.43|52.24|62.91|37.10|51.76|70.64|*41.38*|50.22|
> |CIConv|67.42|70.83|65.19|42.77|61.55|45.40|45.38|57.08|32.44|45.08|
> |CIConv*|**78.38**|*75.67*|*69.33*|55.92|*69.83*|45.75|50.64|*71.93*|35.33|50.91|
> |EGSDG w/ Aug.|*73.67*|72.45|57.31|*57.42*|65.21|*54.11*|53.41|62.74|32.86|50.78|
> |EGSDG w/o Aug.|73.45|**78.48**|**71.94**|**60.13**|**71.00** |**55.14**|*57.34*|**72.50**|**45.84**|**57.71**|

---

> ### Comment · Reviewer_CChP · 2024-03-17
> **The brain tumor segmentation setting is **incorrect**. Please double check.**
>
> Let me explain what might be wrong in a detailed fashion.
> There are three sub-structures in BraTS ("enhancing tumor" (ET), the "tumor core" (TC), and the "whole tumor" (WT) ).
>
> **First, it is not clear which structure to segment**. Were the three sub-structures merged into one label? or individual labels were segmented? No information is given in the main text and the appendix G.
>
> **Second, in BraTS, different modalities see different structures.** For cardiac CT --> MRI, it is fine because the structures are visible and have the same semantic meaning in both modalities. However, in the BraTS case, I would not expect a segmentation method that segments a whole tumor in a T2 scan can segment the whole tumor in T1-c, just because it is not well visible. I would not expect it will segment the enhancing tumor in T1-c either because addressing label shift is not the scope of domain generalization.

---

> > ### Author Response · Authors · 2024-03-18
> >
> > Sorry, we had a misunderstanding with the comment. For BraTS, the mask of each case is shared with four sequences. That is, the masks are the same for different sequences. The file directory of one case in BraTS is as follows:
> > ``` bash
> > BraTS19_TMC_30014_1/
> > ├── BraTS19_TMC_30014_1_flair.nii.gz
> > ├── BraTS19_TMC_30014_1_seg.nii.gz
> > ├── BraTS19_TMC_30014_1_t1ce.nii.gz
> > ├── BraTS19_TMC_30014_1_t1.nii.gz
> > └── BraTS19_TMC_30014_1_t2.nii.gz
> > ```
> > The experimental setup of BraTS is consistent with SADN.
> >
> >
> > - The three sub-structures were merged into one label for segmentation.
> >
> > - The texture and intensity of brain tumors are diverse in different sequences. Therefore, we believe it is a domain generalization problem.

---

> > > ### Comment · Reviewer_CChP · 2024-03-18
> > > **further clarification**
> > >
> > > For BraTS, if you generalize the model trained on 'TMC' (which is the center ID from the example you show) to other center IDs, such as 'Washington', then I am fine with this setup.
> > > What you propose, i.e., training on T2 with a merged tumor map, and generalizing to FLAIR and T1-, does not make sense to me **without defining what you wish to generalize**.
> > >
> > > Maybe I could explain what is a good generalization setup from my understanding:
> > >
> > > 1. acquisition shift: prostate, one site --> other sites (what you presented). This is a common setup. I fully agree with this.
> > >
> > > 2. cross-modality: cardiac CT --> MRI. This also makes sense, because one could see the same anatomy in two modalities.
> > >
> > > However, generalization from T2 to T1-c is different from 2. **There is a label shift in between**. **The shared label map is annotated based on multiple modalities,** i.e., you might not be able to see and segment the merged tumor mask in T1-c because different modalities capture different structures.

---

> > > > ### Author Response · Authors · 2024-03-18
> > > >
> > > > Thank you for the detailed and constructive comments. We admit that there are limitations in the experimental setup of BraTS. Except for the BraTS experiments, we believe that we made a comprehensive analysis to the impact of image edge on the model generalization ability and found that employing the image edge as input to train the model directly brings tremendous generalization performance gains. Extensive experiments on the MMWHS and Prostate datasets demonstrate our method achieves superior generalization performance.

---

> ### Comment · Reviewer_CChP · 2024-03-17
> **there is no adaptation in domain generalization. please use 'deploy'.**
>
> I would agree if you say 'deploy anywhere'. There is no adaptation in domain generalization.
> It is just **technically** confusing if you agree.

---

> > ### Author Response · Authors · 2024-03-18
> >
> > "Train Once, Deploy Anywhere" is a good choice. Thanks again.

---

> ### Comment · Reviewer_CChP · 2024-03-18
> **thank you**
>
> Thank you. Please update the PDF accordingly if it is still possible.
> I raised my score to borderline accept.

---

> > ### Author Response · Authors · 2024-03-22
> >
> > We revised the paper based on the useful suggestions with the following modifications:
> > - We took the "Train Once, Deploy Anywhere" as the title.
> > - We moved Table 3 and Table 4 to the appendix and placed the impact of fusion strategies in the main content.
> > - We discussed the limitations of our model and admitted the experimental limitations of BraTS in the discussion section.
> > - We clarified the performance difference of different fusion strategies and edge detectors.
> >
> > Thanks a lot again.

---

> > > ### Comment · Reviewer_CChP · 2024-03-23
> > > **thank you Jun**
> > >
> > > Thank you for being responsive and updating the manuscript.
> > > My concerns are all addressed.
> > > The score is further updated.

---

### Official Review · Reviewer_gHSU · 2024-02-28

**Confidence:** 4
**Preliminary Rating:** 4
**Recommendation:** Poster
**Final Rating:** 4

**Summary:**

The author present a single-source domain generalization method by using the edge map as the invariant feature for different target distributions. This work includes comprehensive experiments conducted on BraTS, prostate and MMWHS datasets indicating that the proposed method exhibit superior performance comparing with commonly used DG approaches.

**Strengths:**

1. The proposed method is very easy to implement as it only requires a pre-defined data sugmentation method and an edge detector.
2. All kinds of scenarios are thoroughly discussed in the experiment.
3. The performance of the proposed method look promising.

**Weaknesses:**

1. Utilizing the edge information is well-explored in many studys and thus lack of novelty. For example:
> * [Domain generalization for robust MS lesion segmentation](https://www.spiedigitallibrary.org/conference-proceedings-of-spie/12464/124640Z/Domain-generalization-for-robust-MS-lesion-segmentation/10.1117/12.2654373.short)
> * [Transfer Learning with Edge Attention for Prostate MRI Segmentation](https://arxiv.org/pdf/1912.09847.pdf)
> * [Edge U-Net: Brain tumor segmentation using MRI based on deep U-Net model with boundary information](https://www.sciencedirect.com/science/article/pii/S0957417422018516?casa_token=_wEAByImOTkAAAAA:153Iz_CWZKYQrE2WNLIcGReUeXgaqrFACshLAS9ZDIpPAKyvQt_mLRJ8EgSpSffOgURuB8vs9g)

2. The qualitative results in Figure. 2 are very hard to see. Visually, it seems like CSDG and CIConv have similar output with the proposed method.

**Detailed Comments:**

1. Enlarge the qualitative result presented in Figure.2.

2. Discuss more about why some edge detector works better on certain domains. And what is the strategy of selecting it for unseen target domains.

**Justification Of Final Rating:**

The response given by the author coincide with the general intuition. Although the limitations of the proposed method are obvious, they are clearly stated and still worth discussion for future improvement. Thus, I would keep the score as weak accept.

**Justification Of The Preliminary Rating:**

The proposed work is lack of novelty in general. However, the method is very easy to implement and have decent result according to the presented conprehensive experiment. Therefore, I think it is still worth to discuss in MIDL.

**Questions To Address In The Rebuttal:**

1. In the ablation study, the experiment indicate that the selection of edge detector can result in at most 7% of difference in the outcome. It seems that Sobel work well on some of the modalities while AutoCanny does well on others. How to determine which one to use given an unseen domain?

2. The experiment are conducted to segment large targets in general. Therefore, although there is not pre-processing step (e.g., anisotropic diffusion) taken to smooth the image, the noisy edge map doesn't seem to be a problem. But what if the target is small, for example, multiple sclerosis (MS) and cochlear?

**Special Issue:**

No

---

> ### Author Response · Authors · 2024-03-14
>
> Thanks for your useful comments sincerely.
> - Yes, there are indeed many works related to Edge-guided models. For SDG, we note that (i) the medical image segmentation applications where generalization errors often come from imprecise predictions at the ambiguous boundary of anatomies and (ii) the edge of the image is domain-invariant, which can reduce the domain shift between the source and target domain in all network layers. Based on the above motivations, we made a comprehensive analysis to the impact of image edge on the model generalization ability and found that employing the image edge as input to train the model directly brings tremendous generalization performance gains. To the best of our knowledge, we make a first attempt to employ the edge of images as input to train the network. Additionally. due to length limitations, we combined the two images, and the enlarged qualitative result will be added to the Appendix.
>
> - Different edge extractors are sensitive to specific types of edges like edge orientation, noise environment, and edge structure. In different segmentation scenarios, the texture, intensity, and noise of medical images are diverse. This leads to the discrepancy between the valuable edge extraction and the irrelevant noise filtering by different edge extractors, which makes the optimal edge detector on each domain different. The main limitation of our model: On different data sets, the optimal edge detection algorithm is different, which will bring great challenges to choosing the best one for an unseen dataset. We will consider employing the deep learning-based edge detector in an end-to-end fashion in future work.
>
> - Yes, the model has the following limitations:
>     1. The optimal edge extractor is diverse in different segmentation scenarios, which brings great challenges to choosing the best one for an unseen dataset.
>     2. In low-contrast images, the model cannot extract valuable edge information well, which may lead to poor segmentation performance.
>     3. When the segmentation target is small, the extracted edge information may be similar to the surrounding noise, which may lead to the wrong segmentation of the target and noise.

---

> > ### Comment · Reviewer_gHSU · 2024-03-27
> > **Thanks for the response to my questions**
> >
> > 1.I think it is fine to have certain limitations for a specific method and I do recommend to clearly state the limitations to the input image in the conclusion.
> > 2.Maybe using some kinds of ensemble to multiple edge maps acquired by different edge detectors can provide a better outcome.

---

> > > ### Author Response · Authors · 2024-03-28
> > >
> > > Thanks a lot again.
> > > - We clarified the limitations of our model in the discussion section.
> > > - The ensemble strategy is a good idea and we will further explore it in future work.

---

### Official Review · Reviewer_UxTP · 2024-03-04

**Confidence:** 4
**Preliminary Rating:** 3
**Final Rating:** 4

**Summary:**

In this work, the authors tackle the issue of building a segmentation model that, having trained on a single source domain, can generalize well to different kinds of domain shift which would be plausible at test time. The methodology they offer is to pre-process input images with edge detection mechanisms, with the hope that although the image content can change between different domains the edges are consistent features that will transfer better. The authors also discuss how one could benefit from special data augmentation using Bezier curves. Finally, they evaluate all of their claims across three different medical image segmentation datasets, and compare against state of the art SDG models.

**Strengths:**

- The proposed methodology is elegant and simplifies the overcomplicated "learning of shapes" that existing methods use. I buy the argument that this saves time during training, because existing edge detection modules are probably already good enough.
- The authors conduct an extensive amount of experiments with respect to how their method performs under domain shift, and demonstrate in thorough ablations the effect of using different augmentations or edge detectors.
- The proposed methodology is not only simple, and makes a great deal of intuitive sense, but seems to consistently provide gains in Dice score across the different domain shift experiments.

**Weaknesses:**

- It's unclear that the premise of "train once, adapt anywhere" is useful in practice, and if it is, then the datasets chosen are all datasets which have an ample amount of public availability that could be used simultaneously as leverage to train models. I think that the results would be more compelling if the chosen tasks required such an extreme environment as SDG.
- The role of data augmentation/Bezier curves in this paper is confusing. From my perspective, they should be complementary methods to the Edge detectors chosen. Additionally, it's unclear if the other SDG methods were provided the same level of data augmentation (excluding the note on CIConv*). The data augmentation is not mentioned in the contributions but is a large part of the paper.
- All reported dice numbers are lacking variance statistics to quantify how close the reported results are. This is problematic because the authors claim in the contributions that their method achieves superior generalization performance in comparison to baselines, but visually from Figure 2, CSDG, CIConv, and the proposed method all look very similar, and the performance in the table is also very similar.
- One of the contributions listed is explaining that multiple attempts were made of integrating the edge information into the network; however, this result is buried in the appendix and it is unclear why it is a main contribution.
- The writing is not very polished and has both typos and confusing wording.

**Detailed Comments:**

The beginning of section 3.1 has quite a few typos, to list a few.
- Edge detector -> Edge detectors
- filters -> filter
- confusing wording with phrase 'certain types of edges'

**Justification Of Final Rating:**

The authors improved the rigor of their experimental description, including additional information regarding statistical significance, and justified the choices they made. Thus, I believe this paper should be included in the conference as it is a simple method that demonstrates strong results for the proposed task.

**Justification Of The Preliminary Rating:**

My current rating is based on what I believe to be weak explanation for experimental setup and explanation of results. I believe that the proposed method is a valuable contribution and that it makes intuitive sense, but that the inconsistencies with data augmentation being applied and the lack of variance numbers to demonstrate significance leave me giving this paper a borderline rating. If the authors address the way that other SDG methods were given data augmentation and throughout the paper variance numbers were provided for Dice scores and the same story holds, I would consider increasing my score.

**Questions To Address In The Rebuttal:**

Here are a series of questions I had throughout the paper:
- Why pre-make the dataset corresponding to the Bezier curves and not do that on the fly?
- How does EGSDG perform without data augmentation? It seems unfair to compare the traditional SDG methods with EGSDG which was able to train with additional data aug.
- How does EGSDG perform with other kinds of data augmentations that are listed in Table 2?

**Special Issue:**

No

---

> ### Author Response · Authors · 2024-03-14
>
> Thanks for your useful comments sincerely.
> - For medical image segmentation, unsupervised domain adaptation models require retraining when receiving samples from a new data distribution, and multi-source domain generalization methods might be infeasible when there is only a single source domain. These will pose formidable obstacles to model deployment. To this end, we take the "Train Once, Adapt Anywhere" as our objective and consider a challenging but practical problem: Single-source Domain Generalization. In addition, our experimental setup is consistent with existing SDG methods (BraTS[1], Prostate[2], MMWHS[1]).
> > [1] Zhou Z, Qi L, Yang X, et al. Generalizable cross-modality medical image segmentation via style augmentation and dual normalization[C]//CVPR. 2022:20856-20865.
> > [2] Ouyang C, Chen C, Li S, et al. Causality-inspired single-source domain generalization for medical image segmentation[J]. TMI, 2022, 42(4):1095-1106.
>
> - Sorry, we didn't have a reasonable design for the data augmentation experiment. Different augmentation methods should be compared based on the Edge-guided model. The results of Edge-guided models with BézierCurve and RandConv are reported in Table 3, 4, and 9. For Edge-guided models with BasicAug and Cutout augmentation, we found that they were unable to improve the generalization ability after training the model several times (Due to character limitations, the experimental results are not provided). For BézierCurve augmentation, it can indeed be performed on the fly. However, conducting it in the preprocessing stage can effectively reduce the model training time.
>
> - The comparison results with variance are as follows:
> |Methods|T1|T1ce|Flair|Average|
> |-|-|-|-|-|
> |UpperBound|74.42(24.89)|71.64(27.47)|82.75(20.39)|76.27|
> |LowerBound|13.82(14.28)|11.58(17.97)|66.61(30.21)|30.67|
> |IBN-Net|34.37(12.08)|48.27(10.59)|42.33(14.90)|41.66|
> |SW|31.83(15.08)|40.48(17.49)|34.95(19.32)|35.75|
> |RobustNet|8.59(14.10)|10.14(19.81)|68.29(28.76)|29.01|
> |SADN|49.36|38.09|75.87|54.44|
> |CSDG|46.76(31.58)|44.99(32.93)|60.20(31.33)|50.65|
> |CIConv|15.36(15.92)|20.83(19.11)|76.07(28.09)|37.42|
> |CIConv*|53.82(31.75)|52.69(33.42)|74.05(30.53)|60.19|
> |EGSDG w/o Aug.|51.38(32.11)|50.35(32.53)|71.63(29.04)|57.79|
> |EGSDG w/ Aug.|62.59(29.00)|54.68(31.75)|77.07(27.81)|**64.78**|
>
> |Methods|SiteA|SiteC|SiteD|SiteF|SiteE|Average|
> |-|-|-|-|-|-|-|
> |UpperBound|89.13(6.88)|89.96(4.66)|89.31(6.36)|87.76(17.60)|89.34(8.07)|89.10|
> |LowerBound|63.62(21.93)|19.42(28.28)|81.06(20.83)|83.89(27.65)|71.17(33.43)|63.83|
> |IBN-Net|67.36(23.35)|46.79(24.42)|65.09(17.47)|71.45(18.47)|76.88(26.72)|65.51|
> |SW|70.83(21.06)|51.71(22.23)|70.89(19.95)|51.96(24.25)|68.97(19.27)|62.87|
> |RobustNet|73.27(20.27)|55.04(27.54)|77.41(22.07)|54.79(26.18)|70.21(23.07)|66.14|
> |CSDG|69.75(19.50)|61.47(13.95)|74.27(16.42)|76.31(16.54)|70.54(14.71)|70.47|
> |CIConv|73.48(23.32)|63.51(23.18)|80.8(22.88)|62.15(31.14)|74.93(23.04)|70.97|
> |CIConv*|76.41(17.98)|59.74(19.52)|76.63(27.26)|78.10(26.48)|77.17(27.14)|73.61|
> |EGSDG w/o Aug.|72.70(23.52)|59.54(27.28)|83.00(15.74)|70.36(22.73)|81.11(23.45)|73.34|
> |EGSDG w/ Aug.|78.51(22.34)|64.16(23.31)|82.95(16.82)|77.34(19.55)|78.20(18.83)|**76.23**|
>
> |Methods|AA|LAC|LVC|MYO|Average|AA|LAC|LVC|MYO|Average|
> |-|-|-|-|-|-|-|-|-|-|-|
> |UpperBound|89.74(25.06)|84.99(23.12)|87.44(23.29)|83.34(17.38)|86.38|80.76(21.71)|82.29(18.40)|92.38(20.53)|78.23(14.73)|83.42|
> |LowerBound|32.18(3.94)|35.92(9.48)|19.53(6.42)|9.42(2.43)|24.26|18.44(4.91)|8.84(15.87)|38.72(1.32)|9.65(3.10)|18.91|
> |IBN-Net|59.04(29.82)|67.63(26.67)|67.34(26.56)|45.49(27.35)|59.88|31.23(17.55)|42.36(12.36)|59.91(22.43)|34.63(19.99)|42.03|
> |SW|52.94(29.46)|69.52(27.70)|64.28(25.81)|44.64(27.16)|57.84|38.95(21.79)|47.62(15.42)|62.82(23.13)|33.30(19.98)|45.67|
> |RobustNet|68.07(26.80)|74.68(29.06)|62.56(25.75)|46.09(29.11)|62.85|52.27(18.19)|60.08(20.63)|67.26(25.93)|32.97(20.99)|53.14|
> |SADN|51.42|50.20|52.86|52.31|51.70|33.38|31.65|33.29|30.45|32.19|
> |CSDG|66.91(28.00)|68.06(25.69)|64.43(27.66)|52.24(22.37)|62.91|37.10(20.33)|51.76(26.41)|70.64(27.48)|41.38(23.83)|50.22|
> |CIConv|67.42(25.52)|70.83(26.11)|65.19(26.55)|42.77(18.94)|61.55|45.40(21.03)|45.38(25.41)|57.08(26.53)|32.44(15.54)|45.08|
> |CIConv*|78.38(30.61)|75.67(23.37)|69.33(24.96)|55.92(22.92)|69.83|45.75(26.85)|50.64(23.59)|71.93(27.17)|35.33(17.33)|50.91|
> |EGSDGw/oAug.|73.67(29.63)|72.45(28.74)|57.31(28.52)|57.42(24.43)|65.21|54.11(21.11)|53.41(22.31)|62.74(28.22)|32.86(22.27)|50.78|
> |EGSDGw/Aug.|73.45(31.06)|78.48(28.14)|71.94(28.76)|60.13(25.28)|**71.00**|55.14(19.36)|57.34(27.14)|72.50(25.87)|45.84(20.36)|**57.71**|
>
> - Sorry, we didn't well organize the paper. As the main contribution of this work, the analysis of edge fusion strategies should be discussed in the body. In the final version, we moved Table 3 and Table 4 to the appendix and placed the impact of fusion strategies in the main content. In addition, we will double-check the wording of the entire article.

---

> > ### Comment · Reviewer_UxTP · 2024-03-18
> > **Further clarification on questions.**
> >
> > Thank you to the authors for the response, here are a few things I still am wondering.
> >
> > 1. For data augmentation, what I was more interested in knowing was how the proposed baselines methods, not the proposed methods, did under the different data augmentation regimes because it seems like when CIConv was given the bezier curves dataset that it became the best performer. I would like to know if applying even the basic kinds of augs during the training process of the other methods gave similar boosts.
> >
> > 2. The variance numbers leave me very confused as to the statistical significance of the improvement over other baselines methods. I think it would be better to show 95% confidence intervals of the statistics and (if possible) to run paired-t tests across the results to a) know when things are actually significant and b) to co-bold things that perform the same.
> >
> > For the time being I will maintain my score of borderline, but I am open to increasing it if the proposed results end up in fact being statistically significant.

---

> ### Author Response · Authors · 2024-03-22
>
> Thank you for the detailed clarification.
>
> - The results of baseline methods with data augmentation are as follows (there are augmentation operations in RobustNet and CSDG):
> ## BraTS
> |Methods|T1|T1ce|Flair|Average|
> |-|-|-|-|-|
> |IBN-Net_BasicAug|13.53|20.58|68.92|34.34|
> |SW_BasicAug|15.34|23.99|68.43|35.92|
> |IBN-Net_BezierCurve|48.35|44.80|64.28|52.48|
> |SW_BezierCurve|42.95|42.09|57.20|47.41|
>
>
> ## Prostate
> |Methods|SiteA|SiteC|SiteD|SiteF|SiteE|Average|
> |-|-|-|-|-|-|-|
> |IBN-Net_BasicAug|73.39|57.37|69.65|60.14|75.27|67.16|
> |SW_BasicAug|71.67|60.39|70.88|55.67|73.00|66.32|
> |IBN-Net_BezierCurve|68.57|44.05|65.70|68.16|74.12|64.12|
> |SW_BezierCurve|71.49|57.71|74.02|62.47|73.93|67.92|
>
> ## MWHS
> |Methods|AA|LAC|LVC|MYO|Average|
> |-|-|-|-|-|-|
> |IBN-Net_BasicAug|66.22|69.55|71.19|56.99|65.99|
> |SW_BasicAug|61.48|72.71|70.20|54.40|64.70|
> |IBN-Net_BezierCurve|61.41|59.19|69.37|50.19|60.04|
> |SW_BezierCurve|50.46|54.61|61.03|46.34|53.11|
>
> - The comparison results with 95% confidence intervals are as follows (we admit that there are discriminatory limitations of the selected samples for different methods in the qualitative results)：
> ## BraTS
> |Methods|T1|T1ce|Flair|Average|
> |-|-|-|-|-|
> |UpperBound|74.42(72.85,75.98)|71.64(69.90,73.37)|82.75(81.57,83.92)|76.27|
> |LowerBound|13.82(12.92,14.71)|11.58(10.44,12.71)|66.61(64.86,68.35)|30.67|
> |IBN-Net|34.37(33.61,35.12)|48.27(47.60,48.93)|42.33(41.46,43.19)|41.66|
> |SW|31.83(30.43,32.32)|40.48(39.37,41.58)|34.95(33.83,36.06)|35.75|
> |RobustNet|8.59(7.70,9.47)|10.14(8.89,11.38)|68.29(66.62,69.95)|29.01|
> |CSDG|46.76(44.77,48.74)|44.99(42.91,47.06)|60.20(58.38,62.01)|50.65|
> |CIConv|15.36(14.35,16.36)|20.83(19.62,22.03)|76.07(74.44,77.69)|37.42|
> |CIConv*|53.82(51.82,55.81)|52.69(50.24,54.45)|74.05(72.28,75.81)|60.19|
> |EGSDGw/oAug.|51.38(49.36,53.39)|50.35(48.29,52.40)|71.63(69.95,73.30)|57.79|
> |EGSDGw/Aug.|62.59(60.76,64.41)|54.68(52.67,56.68)|77.07(75.46,78.67)|**64.78**|
>
> ## Prostate
> |Methods|SiteA|SiteC|SiteD|SiteF|SiteE|Average|
> |-|-|-|-|-|-|-|
> |UpperBound|89.13(87.64,90.61)|89.96(88.88,91.03)|89.31(86.76,91.85)|87.76(82.30,93.21)|89.34(86.17,92.50)|89.10|
> |LowerBound|63.62(58.90,68.33)|19.42(12.88,25.95)|81.06(72.72,89.39)|83.89(75.32,92.45)|71.17(58.06,84.27)|63.83|
> |IBN-Net|67.36(62.33,72.38)|46.79(41.14,52.43)|65.09(58.10,72.07)|71.45(65.72,77.17)|76.88(66.40,87.35)|65.51|
> |SW|70.83(66.29,75.36)|51.71(46.57,56.84)|70.89(62.90,78.87)|51.96(44.44,59.47)|68.97(61.41,76.52)|62.87|
> |RobustNet|73.27(68.90,77.63)|55.04(48.67,61.40)|77.41(68.58,86.23)|54.79(46.67,62.90)|70.21(61.16,79.25)|66.14|
> |CSDG|69.75(65.55,73.94)|61.47(58.24,64.69)|74.27(67.70,80.83)|76.31(71.18,81.43)|70.54(64.77,76.30)|70.47|
> |CIConv|73.48(68.46,78.49)|63.51(57.15,68.86)|80.8(71.64,89.95)|62.15(52.49,71.80)|74.93(65.89,83.96)|70.97|
> |CIConv*|76.41(72.54,80.27)|59.74(55.23,65.84)|76.63(65.72,87.53)|78.10(69.89,86.30)|77.17(66.53,87.80)|73.61|
> |EGSDGw/oAug.|72.70(67.63,77.76)|59.54(54.23,65.84)|83.00(76.70,89.29)|70.36(63.31,77.40)|81.11(71.91,90.30)|73.34|
> |EGSDGw/Aug.|78.51(73.70,83.31)|64.16(58.77,69.54)|82.95(76.22,89.679)|77.34(71.28,83.40)|78.20(70.81,85.58)|**76.23**|
>
> ## MMWHS
> |Methods|AA|LAC|LVC|MYO|Average|AA|LAC|LVC|MYO|Average|
> |-|-|-|-|-|-|-|-|-|-|-|
> |UpperBound|89.74(87.49,91.98)|84.99(82.91,87.06)|87.44(85.35,89.52)|83.34(81.78,84.89)|86.38|80.76(79.66,81.85)|82.29(81.26,83.11)|92.38(91.34,93.41)|78.23(77.48,78.97)|83.42|
> |LowerBound|32.18(31.82,32.53)|35.92(35.07,36.76)|19.53(18.95,20.10)|9.42(9.20,9.63)|24.26|18.44(18.19,18.68)|8.84(8.04,9.63)|38.72(38.65,38.78)|9.65(9.49,9.80)|18.91|
> |IBN-Net|59.04(56.36,61.71)|67.63(65.23,70.02)|67.34(65.04,69.63)|45.49(43.03,47.94)|59.88|31.23(30.34,32.11)|42.36(41.73,42.98)|59.91(58.78,61.03)|34.63(33.62,35.63)|42.03|
> |SW|52.94(50.29,55.58)|69.52(67.03,72.00)|64.28(61.96,66.59)|44.64(42.20,47.07)|57.84|38.95(37.85,40.04)|47.62(46.84,48.39)|62.82(61.65,63.98)|33.30(32.29,34.30)|45.67|
> |RobustNet|68.07(65.66,70.47)|74.68(72.07,77.28)|62.56(60.25,64.86)|46.09(43.48,48.69)|62.85|52.27(51.35,53.18)|60.08(59.04,61.11)|67.26(65.95,68.56)|32.97(31.91,34.02)|53.14|
> |CSDG|66.91(64.39,69.42)|68.06(65.75,70.36)|64.43(61.95,66.90)|52.24(50.23,54.24)|62.91|37.10(36.07,38.12)|51.76(50.43,53.08)|70.64(69.25,72.02)|41.38(40.18,42.57)|50.22|
> |CIConv|67.42(65.13,69.70)|70.83(68.48,73.17)|65.19(62.80,67.57)|42.77(41.07,44.46)|61.55|45.40(44.34,46.45)|45.38(44.10,46.65)|57.08(55.74,58.41)|32.44(31.65,33.22)|45.08|
> |CIConv*|78.38(75.63,81.12)|75.67(73.57,77.76)|69.33(67.09,71.56)|55.92(53.86,57.97)|69.83|45.75(44.40,47.09)|50.64(49.45,51.82)|71.93(70.56,73.29)|35.33(34.45,36.20)|50.91|
> |EGSDGw/oAug.|73.67(71.01,76.32)|72.45(69.87,75.02)|57.31(54.73,59.88)|57.42(55.22,59.61)|65.21|54.11(53.04,55.17)|53.41(52.28,54.53)|62.74(61.32,64.15)|32.86(31.74,33.97)|50.78|
> |EGSDGw/Aug.|73.45(70.66,76.23)|78.48(75.95,81.00)|71.94(69.36,74.51)|60.13(57.86,62.39)|**71.00**|55.14(54.16,56.11)|57.34(55.97,58.70)|72.50(71.19,73.80)|45.84(44.81,46.86)|**57.71**|

---

> > ### Comment · Reviewer_UxTP · 2024-03-26
> > **Thank you to the authors.**
> >
> > Thank you for the thorough addition of 95% confidence intervals. I think that these show that EGSDG w/ Aug indeed shows a thorough improvement over baselines (and thus should be included in the paper), and thus I am changing my score to accept.

---

> > > ### Author Response · Authors · 2024-03-27
> > >
> > > Thanks a lot again. We revised the paper based on the useful suggestions.

---

### Official Review · Reviewer_hAg2 · 2024-03-05

**Confidence:** 4
**Preliminary Rating:** 4

**Summary:**

The authors proposed a method for single-source domain generalization. Rather than directly using the images as inputs, the authors proposed to use the edge maps as the network input. The results show that training solely on the edge maps could outperform the competing methods that are trained on images.

**Strengths:**

The paper is well written and many competing methods have been compared against the proposed one. It is also interesting to see how using the edge maps along could achieve superior performance for single-source domain generalization.

**Weaknesses:**

The results of Table 3 and Table 4 are not informative. The ablation studies are showing that with Bezier curve (intensity flipping) augmentation the results can be improved. This is expected as random contrast adjustment has been one of the most important augmentation techniques for quite a while. The Bezier curve augmentation may be simply replaced by gamma adjustment, as long as the augmentation can help flip the intensity profiles. Instead, I believe the experiments in the appendix about fusion strategies should be moved to the main content. This is because the authors are claiming that using edge map directly as input, rather than late fusion or edge + image, is a better strategy. This is one of the major findings of this paper and thus I would recommend to move these experiments to the main paper.

**Detailed Comments:**

See weakness section. It would be helpful to further clarity the performance difference or impacts of fusion strategies of these edge maps.

**Justification Of The Preliminary Rating:**

The proposed method seems to be interesting and it should be worthy to be exposed to the community for discussions.
Before accepting this paper, I would highly recommend the authors move the Table 3 and 4 to appendix, and move the impact of fusion strategies to the main paper.

**Questions To Address In The Rebuttal:**

N/A

---

> ### Author Response · Authors · 2024-03-14
>
> Thanks for your useful comments sincerely.
> - Yes. For the BraTS dataset, we note that the tumor is brighter in the T2 sequence and darker in the T1 sequence. Therefore, the intensity flipping augmentation can significantly improve the generalization performance. However, taking only intensity flipping may hurt the model's segmentation accuracy on other datasets, like the cross-center Prostate dataset.
>
> - Sorry, we didn't well organize the paper. As the main contribution of this work, the analysis of edge fusion strategies should be discussed in the body. In the final version, we moved Table 3 and Table 4 to the appendix and placed the impact of fusion strategies in the main content.
>
> - These edge-guided strategies can be divided into two categories:
>     1. using the edge map as an additional supervision signal to increase the model’s attention at the boundary of segmentation targets. The performance is lower than ours, probably because this category strategy weakens the supervision of the edge information via the segmentation head at the training stage.
>     2. concatenating the image or its features with the edge map to force the model learning domain invariant representation and enhance the generalization ability. The performance is lower than ours, possibly due to the grayscale information making the model overfit on the source domain.

---

### Meta-Review · Area_Chair_ouiC · 2024-04-04

**Recommendation:** Accept (Poster)
**Confidence:** 4

**Metareview:**

The paper  proposes a new domain generalization/adaptation technique for medical image segmentation.

The original reviews appreciated the work, but raised concerns of clarity and claims and experimental setup, However, after a productive rebuttal and discussion period, The authors were able to adapt the work and improve the experimental presentation, and the reviewers all agree that the paper should be accepted.

Congratulations to the authors and I appreciate the thorough review and discussion from the reviews!

---

### Decision · Program_Chairs · 2024-04-06

Accept (Poster)